# Winter outdoor thermal comfort of older people with different degrees of frailty: A study in Nanyang city, China's hot-summer and cold-winter region

Dong Yan[1], Peng Zhang[2], Zhiyuan Liu[1], Ran Chen[3]*, Biao Wang ORCID[4]*

**1** Architecture College, Nanyang Institute of Technology, Nanyang, China, **2** School of Architecture and Engineering, Wuhan City Polytechnic, Wuhan, China, **3** Department of Civil, Environmental and Geomatic Engineering, University College London, London, United Kingdom, **4** School of Architecture, Soochow University, Suzhou, China

* ran.chen.09@ucl.ac.uk (RC), uangning@suda.edu.cn (BW)

## Abstract

The thermal comfort of outdoor environments in winter directly impacts the spatial utilization efficiency of urban parks and the physical and mental health of the elderly. This study uses Nanyang People's Park, which is popular for local elderly, as a case study. Using the FRAIL scale to categorize different levels of frailty, combined with objective microclimate field measurements and subjective thermal comfort questionnaires, the study conducts regression analysis between the calculated physiological equivalent temperature and the thermal comfort voting results. A model for evaluating the thermal comfort of older people in urban parks during winter in hot summer and cold winter regions (HSCW) is established. The study also explores the relationships between physical, personal, social, and psychological factors and the outdoor thermal comfort of the elderly, as well as the differences in thermal adaptation among older people with varying degrees of frailty. The research findings show that the neutral temperature for no frailty is 10.29°C, 9.60°C for pre-frailty, and 8.65°C for severe frailty. As the degree of frailty increases, the outdoor thermal sensitivity of the older people gradually weakens. The adaptive behaviors of older people with different levels of frailty are greatly influenced by their specific environment. The research results provide basic data reference for the design of aging-friendly urban parks in HSCW, and also provide a renovation reference for relevant government departments to formulate the thermal comfort standards of urban parks.

## Introduction

The problem of global population aging has become increasingly significant, with data showing that the number of people aged 60 and over is expected to increase to 20% by 2050 [1]. The situation in China is even more concerning. By 2024, the

**Data availability statement:** All relevant data are within the manuscript and its Supporting information files.

**Funding:** This research was funded by the Science and technology project of Henan Province (252102320315), Housing and Urban Rural Construction Science and Technology Plan Project (HNJS-2024-R1), and Henan Province soft science project (242400410262, 252400410248). The funders had no role in study design, data collection and analysis, decision to publish, or preparation of the manuscript.

**Competing interests:** The authors have declared that no competing interests exist.

**Abbreviations:** BS, The square adjacent to building; EP, Exercise plaza, ES, Entrance square; HSCW, Hot-summer and cold-winter; MTSV, The mean values of TSV; NPET, Neutral temperature; NPETR, The neutral temperature range; OTC, Outdoor thermal comfort; PET, Physiological equivalent temperature; RS, Roadside seats; RH, Relative humidity; SET, Standard effective temperature; SVF, Sky view factor; Ta, Air temperature; Tmrt, Mean radiant temperature; Tg, Black globe temperature; TSV, Thermal sensation vote; TCV, Thermal comfort vote; TAV, Thermal acceptability votes; UTCI, Universal thermal climate index; Va, Wnd speed; ws, The square adjacent to water.

number of people aged 60 and over in China has exceeded 300 million, accounting for about 22% of the total population. This proportion continues to rise, and it is projected that by 2035, the number of older people will surpass 400 million, making up over 30% of the total population, marking the entry into a stage of severe aging. Research has shown that there are significant differences in outdoor thermal comfort between the elderly and non-elderly populations [2–4]. As an important green infrastructure, the effective plant configuration in urban parks plays a good effect of wind and dust washing in winter [5–8]. Therefore, urban parks are not only an important place for the elderly to enjoy outdoor leisure and entertainment, but also a key platform for the supply of elderly care services. Although indoor heating in winter will give the elderly a relatively comfortable environment, it is beneficial and necessary to go out for exercise and communication [9,10]. A comfortable outdoor environment can reduce the energy consumption of equipment, such as the frequency of air conditioning and heating, so as to achieve China's macro goal of "peak carbon emissions before 2030" [11,12].

In China, which is hot in summer and cold in winter, the winter is cold and there are many extreme weather. The climate can not only directly damage human health through the action of a single element, but also increase the risk of disease through the comprehensive action between different elements, thus indirectly leading to population death [10]. The elderly are a sensitive group to climate change, and extreme weather or climate change can have many adverse effects on their health. In Nanyang City, the majority of park visitors are older people. The thermal comfort outdoors in winter will directly impact the spatial utilization efficiency of urban parks and the physical and mental well-being of the elderly. Therefore, creating a suitable thermal comfort micro-environment for the elderly in areas like urban parks is particularly important.

Outdoor Thermal Comfort (OTC) is the key to determining the success or failure of urban park design. Thermal comfort is often influenced by both environmental and individual factors. Environmental factors include temperature (Ta), relative humidity (RH), wind speed (v), and mean radiant temperature (Tmrt). Individual factors include height, weight, age, gender, clothing, and activity. Most studies use software to integrate multiple environmental and individual indices into a single index such as the Physiological Equivalent Temperature (PET) or universal thermal climate index (UTCI), to evaluate outdoor thermal comfort [13,14]. But OTC is defined by Refrigerating and Air-Conditioning Engineers (ASHRAE) as "a state of mind in which one feels satisfied with the environment" [15]. That is, for the same outdoor environment, different types of people will have different thermal sensations. Therefore, it is necessary to conduct thermal sensation voting while measuring the microclimate on the spot. Combining subjective and objective is the mainstream of OTC research at present [16,17]. Most studies have shown a high correlation between the objective environment and subjective thermal sensation, leading to regression equations between PET or UTCI and thermal sensation voting, and then to the comfort threshold applicable to different populations in different climate zones. At present, such studies have covered most parts of the world [18–20]. In recent years, the research

results of OTC in China have increased rapidly, covering different climate zones and different cities in the same climate zone.

Table 1 summarizes the research status of outdoor thermal comfort in hot summer and cold winter areas in China, which includes 15 research results combining field measurement and questionnaire survey.

Currently, most outdoor thermal comfort research focuses on popular provincial capitals, such as Changsha, Wuhan, and Chengdu, leaving prefecture-level cities like Nanyang understudied. However, these smaller cities house significant aging populations and face distinct microclimatic conditions that may differ from major metropolitan areas. Nanyang, located in southwestern Henan Province, is a representative city in China's hot-summer and cold-winter (HSCW) climate zone, characterized by humid, hot summers and cold, dry winters. Its urban morphology, population density, and aging demographics are typical of many medium-sized Chinese cities, making it an ideal case for understanding the thermal comfort needs of older adults beyond provincial capitals. Furthermore, as microclimates and individual variables jointly influence human thermal sensation, research has increasingly moved beyond mixed-population studies to examine population subgroups based on gender, age, and activity [36–38]. Investigating frailty thermal responses in a representative HSCW city like Nanyang addresses a critical research gap and provides insights applicable to similar cities across the region.

In addition to the direct influence of physical environment and individual physiological factors, OTC is also influenced by social and psychological factors [39]. Among them, social factors include occupation, education level, whether there are companions or not, etc. Some studies have shown that people with high education and high income are more sensitive to outdoor environment [40,41]. People with companions feel more comfortable in their environment [42]. Psychological factors include environmental satisfaction, thermal experience, frequency of use, etc. Some studies have shown that the

**Table 1. Summary of OTC studies in China's hot summer and cold winter regions.**

| Year | City | Latitude | Longitude | Type | Season | Survey population | Variable |
|---|---|---|---|---|---|---|---|
| 2009 [21] | Nanjing | 32°14′ | 118°42′ | Campus | Summer | Student | Gender, emotion, movement |
| 2010 [22] | Changsha | 28°11′ | 112°58′ | City parks, squares, streets, campuses | Summer | Mixed crowd | – |
| 2010 [23] | Changsha | 28°11′ | 112°58′ | City parks, squares, streets, campuses | Summer | Mixed crowd | Gender |
| 2011 [24] | Wuhan | 30°35′ | 114°17′ | Square | Summer | Mixed crowd | Sunshade |
| 2012-2013 [25] | Changsha | 28°11′ | 112°58′ | City streets, campuses | Annual | Mixed crowd | – |
| 2011-2014 [26] | Wuhan | 30°35′ | 114°17′ | Avenue | Annual | Mixed crowd | Gender, age, activity |
| 2011 [27] | Wuhan | 30°35′ | 114°17′ | Community outdoor space | Summer, autumn | Mixed crowd | – |
| 2012 [28] | Chengdu | 30°39′ | 104°3′ | pedestrian mall | Summer | Mixed crowd | – |
| 2014-2015 [29] | Shanghai | 30°40′—31°53′ | 120°51′—122°12′ | City park | Autumn, winter | Mixed crowd | Short-term experience, gender, age, length of residence |
| 2018 [30] | Mianyang | 31°48′ | 104°73′ | Campus | Summer, winter | Student | Gender, activity, shade |
| 2018 [31] | Mianyang | 31°48′ | 104°73′ | City park | Summer, autumn, winter | Mixed crowd | – |
| 2019 [32] | Chengdu | 30°39′ | 104°3′ | Campus | Summer, winter | Mixed crowd | Landscape type |
| 2019 [33] | Chengdu | 30°39′ | 104°3′ | City park | Summer, winter | Mixed crowd | Landscape type |
| 2021 [34] | Hainning | 30°27′ | 120°34′ | Outside the factory | Winter | Mixed crowd | architectural environment |
| 2023 [35] | Mount Huang | 29°43′ | 118°19′ | City park | Winter | Old people | Age, intensity and content of activity |

neutral PET in different cities is related to the background urban climate due to the influence of thermal experience [43]. The longer the residence time, the stronger the ability to adapt to the environment, and the higher the acceptance of the environment [20,41].

In addition, according to the existing studies, the focus of researchers is still on mixed populations, and outdoor thermal comfort studies for special groups such as the elderly or children are rarely seen [44,45]. There are significant differences in outdoor thermal sensation between the elderly and non-elderly. At present, a few research results have classified the elderly by age group to explore the differences in outdoor thermal sensation among the elderly of different age groups [35,40,46,47]. However, age based grouping is essentially a division based on the "time dimension", which assumes homogeneity of physical function among the elderly within the same age group. However, age itself cannot accurately reflect the true degree of physiological decline (i.e., frailty) of the elderly [48]. In contrast, classification based on the degree of frailty is a classification based on the "health dimension". It directly focuses on the differences in stress resistance among the elderly due to decreased physiological reserves and multiple system dysfunction. The scientific value of this classification method lies in its breaking away from the traditional research perspective of treating the elderly as a "homogeneous group" under age grouping, and instead recognizing and quantifying the heterogeneity of this group. In fact, there are significant differences in physiological regulation abilities (such as sweating, vasoconstriction) and psychological perception thresholds among elderly people with different degrees of weakness (such as strong, pre weak, and weak), leading to their different responses to changes in outdoor environments. For example, research on indoor spaces in residential buildings in hot summers and cold winters in China has shown that elderly people with different degrees of frailty exhibit different thermal adaptation behaviors in winter [49]; Similarly, in the case of sudden changes in indoor temperature during summer, there are significant differences in subjective feelings and physiological regulation among elderly people with different degrees of frailty [50,51].

Therefore, introducing the concept of "frailty level" into outdoor thermal comfort research has additional scientific value in that it goes beyond the rough division based solely on the time dimension (age) and instead starts from the actual physiological health status of the human body, providing a more explanatory theoretical framework for explaining and predicting the huge individual thermal comfort differences in the elderly population. This will help to develop more refined and inclusive design standards for aging friendly outdoor environments in the future, rather than just serving "statistically significant elderly people".

The degree of physical frailty in the elderly is often influenced by factors such as age, multiple diseases, sleep, and genetics. Currently, four frailty scales used in geriatric medicine are the Fried Frailty Phenotype, the FRAIL Scale, the Edmonton Frailty Assessment Scale, and the Tilburg Frailty Indicator. Among these, the FRAIL Scale used in this study is suitable for rapid large-scale screening. Due to its short assessment time and ease of operation, it does not require specialized equipment, leading to high acceptance and response rates among the elderly [52,53]. Although the above frailty scales are widely used in the medical field to classify the degree of frailty in the elderly, no relevant studies have been found to link the degree of frailty in the elderly with the thermal comfort of outdoor open space [49–51]. In the variable and extreme outdoor environment in winter, whether the difference of thermal perception caused by this degree of frailty still exists, and which factors significantly affect the elderly with different degrees of frailty in outdoor environment are the problems that need to be solved in this study.

This paper obtains four microclimate variables—air temperature, humidity, black globe temperature, and wind speed—at measurement points through objective on-site measurements. It also conducts subjective questionnaire surveys to assess the degree of frailty in older people using the FRAIL Scale and categorizes them accordingly. With the help of Rayman software, PET is calculated to verify the correlation between thermal comfort ratings and PET. A regression equation is established to explore the thermal comfort thresholds for older people with different degrees of frailty and the differences in thermoneutral temperatures. The study establishes a winter thermal comfort evaluation model for older people with different degrees of frailty in urban parks in regions with hot summers and cold winters, while also exploring

the relationships between physical, personal, and psychosocial factors and outdoor thermal comfort in the elderly. This research primarily focuses on changes in OTC among older people with different degrees of frailty, enriching the findings on OTC in China's hot summer and cold winter regions.

The main contributions of this study are:

(1) Establish outdoor heat benchmarks for older people with different degrees of frailty;

(2) Discuss the importance of physical, personal, social and psychological factors to the elderly with different degrees of frailty;

(3) Put forward corresponding design strategies to improve the age-appropriate level of urban parks.

## 2. Methodology

### 2.1 Research area

Nanyang City is located in the southwestern part of Henan Province, China, with a subtropical monsoon climate. It falls within the summer hot and winter cold region according to China's architectural climate zoning. Meteorological data from 2011 to 2021 show that the highest monthly average temperature was 33.1°C, occurring in July, while the lowest monthly average temperature was-1.9°C, in January. The coldest months are January and February, with monthly average temperatures ranging from-1.9°C to 1.3°C. The annual average relative humidity fluctuates between 44% and 86%.

The field measurement site is Nanyang Peoples Park which is located in the central urban area of Nanyang city (1 12°5 4 E, 3 3° 00N), with a total area of about 11.5hm². It is a composite urban park that integrates landscape recreation, pedestrian traffic, and commercial performances. Its excellent location and multifunctional nature attract various older people, making it an ideal place for studying outdoor thermal comfort among older people of different frailty levels. Based on the concentration of older people, spatial types, and functions, six typical spaces were selected as measurement points and categorized into five classes: entrance square (ES), Exercise plaza (EP), the square adjacent to water (WS), the square adjacent to building (BS), and Roadside seats (RS). For on-site surveyed photos of different locations, see Fig 1.

### 2.2 Meteorological measurement

The recruitment period for this study was from 01/12/2024 to 28/02/2025.

The actual meteorological measurement took place on December 14, 2024, December 21, 2024, January 5, 2025, January 25, 2025, February 8, 2025, and February 22, 2025, totaling six days. The selection of these six measurement days was based on the local long-term meteorological records (2011–2021) to ensure representativeness of typical winter conditions in Nanyang City. Specifically, the selected dates cover the coldest period (January) and the transition months (December and February). According to historical data, daily average temperatures during winter typically range from −1.9°C to 8.5°C, and the recorded air temperatures during our measurement days (ranging from −2.9°C to 15.1°C) captured the full spectrum of typical winter weather, including both the lowest monthly averages.

All measurement days were sunny or partly cloudy, representing the most common weather conditions for outdoor activities among the elderly, as overcast or rainy days usually deter them from visiting parks. We intentionally included the Winter Solstice (December 21) as a measurement day. On this day, the sun reaches its lowest solar elevation angle of the year, which results in the weakest solar radiation intensity and the longest shadows. This represents a critical extreme condition for outdoor thermal comfort in winter, allowing us to capture the lower bound of thermal exposure that elderly individuals may experience.

Based on the winter weather conditions and the activity patterns of the older people, the final measurement period was determined to be from 9:00 AM to 5:00 PM each day. Air temperature (Ta), relative humidity (RH), wind speed (Va), and black globe temperature (Tg) were collected every minute at the measurement point. The air temperature (Ta) and relative

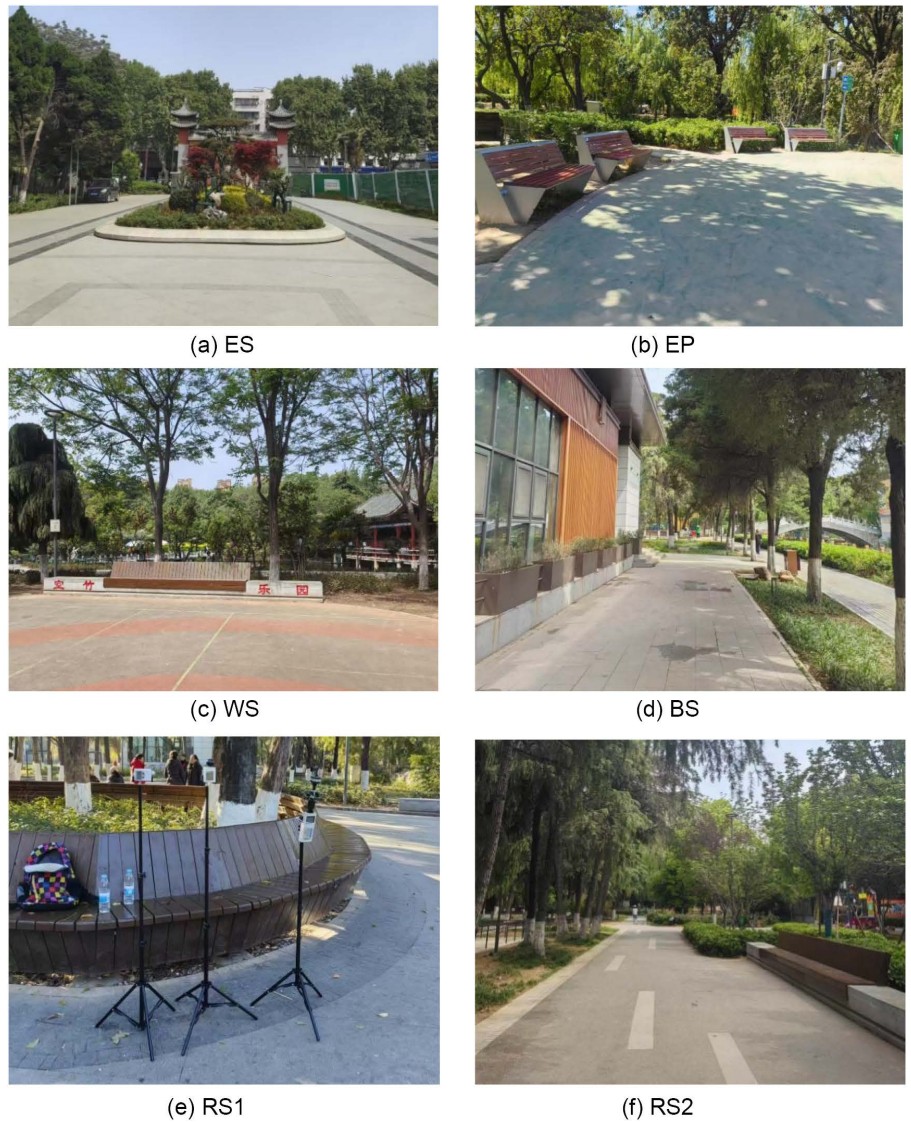

**Fig 1. Surveyed photos of different locations. (a)** ES **(b)** EP **(c)** WS **(d)** BS**(e)** RS1 **(f)** RS2.

humidity (RH) were recorded using a USB type COS-04–0 automatic temperature and humidity recorder, the black globe temperature (Tg) was recorded using a USB type RS/FV-HQ-USB-SJJ black globe temperature recorder, and the wind speed (Va) was recorded using an RS-HSQXZ-USB-2 mechanical handheld weather station. All instruments met the ISO7726 standard requirements for measurement range and accuracy. Specific instrument parameters are listed in Table 2, and all instruments were set up at a height of 1.5 meters above the ground. The average radiative temperature ($T_{mrt}$) was calculated using ISO7726 standards [54]:

$$T_{mrt} = \left[ (T_g + 273.15)^4 + \frac{1.1 \times 10^8 V_a^{0.6}}{\varepsilon D^{0.4}} (T_g - T_a) \right]^{\frac{1}{4}} - 273.15$$

(1)

where D is the globe diameter (D = 0.05 m in this study) and $\varepsilon$ is the emissivity ($\varepsilon$ = 0.95 for a black globe).

**Table 2. Technical parameters of testing instruments.**

| Instrument | Test items | Measuring range | Accuracy |
|---|---|---|---|
| USB type COS-04–0 automatic temperature and humidity recorder | Air temperature | −20°C~60°C | ±0.3°C |
| | Relative humidity | 0~100% | ±2% |
| RS/FV-HQ-USB-SJJ black globe temperature recorder | Black globe temperature | −40°C~120°C | ±0.2°C |
| RS-HSQXZ-USB-2 mechanical hand meteorological station | Wind speed | 0~30m/s | ± (0.3m/s±0.03*v) (v is the true wind speed) |

### 2.3 Questionnaire survey

The study was conducted in accordance with the Declaration of Helsinki, and approved by the Institutional Review Board of Architecture College, Nanyang Institute of Technology (Approval number: NYISTIRB 2025-026, 2024.8). The recruitment period for this study was from 01/12/2024 to 28/02/2025. All participants in the study had provided informed consent prior to enrollment. Trained researchers explained the study objectives, procedures, risks and benefits, confidentiality measures, as well as the rights to voluntary participation and withdrawal in plain language that participants could understand, in person. The consent process was witnessed by an independent observer (Li Yanke, staff member of Nanyang People's Park). The oral consent process was documented by the observer using a standardized "Oral Consent Record Form," which detailed the key points of consent, questions raised by the participant, and the consent statement. After the participant's oral consent, both the observer and the researcher signed the record form for confirmation.

Since China defines people over 60 years old as older people, only people over 60 years old were surveyed, and meteorological parameters were recorded at the same time during the questionnaire survey.

This questionnaire is divided into three parts. The first part covers basic information about the older people, including age, gender, clothing, and activity level, with social and psychological factors added. Social factors include education level, companionship, and monthly income; psychological factors include purpose, frequency of use, and overall satisfaction. The second part investigates the degree of frailty in the older people using the FRAIL scale, consisting of five questions, with scores ranging from 0−1 for no frailty, 2 for pre-frailty, and 3−5 for significant frailty. The FRAIL scale was administered through face-to-face interviews conducted simultaneously with the thermal comfort survey. Each participant was asked the five FRAIL questions verbally by the investigator, and responses were recorded immediately to ensure accuracy and completeness. The third part evaluates thermal comfort, including Thermal Sensation Vote(TSV), Thermal Comfort Vote (TCV), and Thermal Acceptability Votes (TAV). Thermal sensation uses the ASHRAE standard 7-point scale (−3 cold, −2 cool, −1 slightly cool, 0 neutral, +1 slightly warm, +2 warm, +3 hot), thermal comfort uses a 5-point scale (−2 very uncomfortable, −1 uncomfortable, 0 neutral, +1 comfortable, +2 very comfortable), and thermal acceptability uses a 2-point scale (−1 unacceptable, +1 acceptable). For positive evaluations of thermal environment parameters, temperature, humidity, wind speed, and solar radiation are rated using a 3-point scale (−1 decrease, 0 no change, +1 increase) according to the ASHRAE standard. At the end of the questionnaire, respondents are asked to describe their adaptive behaviors when feeling cold or hot, such as adding more clothes or moving towards the sunshine; move to the shade and reduce clothing when feeling hot.

Informed Consent Statement: Informed consent was obtained from all subjects involved in the study. Written informed consent was obtained from the patient(s) to publish this paper.

### 2.4 Evaluation indicators

Thermal comfort evaluation is a quantitative evaluation of human thermal comfort degree using relevant standards. It integrates multiple physical and physiological indexes into a single index for convenience. Common outdoor thermal comfort evaluation indexes are PET, PMV, UTCI and SET. In a review by D. Lai, 53.5% of the relevant studies used the above

evaluation indicators, and PET alone accounted for 30.2% of the usage [20]. It is an index of thermal comfort evaluation for different climates recognized by the German Engineering Association. Therefore, PET is used in this paper as the evaluation index, and the measured meteorological data and personal data are input into Rayman software for calculation. The required input parameters included meteorological data (air temperature, relative humidity, wind speed, and mean radiant temperature derived from global radiation) and personal parameters. Based on the surveyed population characteristics, the following standard assumptions were applied: height (1.65 m, representing the average of the older adult sample), weight (65 kg, based on measured sample average), age (65 years, sample mean), gender (male/female as recorded), clothing insulation (0.9 clo, typical value for the season and survey period).

## 3. Results

### 3.1 Descriptive analysis

(1) Meteorological parameters

Table 3 presents the meteorological conditions recorded at the six measurement points during the field survey. Point ES exhibited the highest average air temperature (Ta) and ground surface temperature (Tg), attributable to its location in an unshaded entrance plaza with hard paving that contributes significant heat radiation to the surrounding air. In contrast, point BS recorded the lowest average Ta and Tg, due to shading from adjacent buildings and a higher density of surrounding trees. During winter, the maximum temperature differential across all points was within 1°C, indicating that peak temperatures remained relatively consistent regardless of measurement location.

The highest relative humidity was recorded at point WS, resulting from its proximity to a water body. Solar radiation promotes water evaporation, increasing moisture content in the surrounding air and creating a humid microenvironment. Beyond WS, relative humidity levels at the other five points showed minimal variation. Regarding wind speed, WS also exhibited the highest maximum wind speed (3.0 m/s), while variations among the remaining points were negligible. These findings demonstrate how differences in location and spatial configuration among measurement points generate discernible variations in microclimatic conditions.

(2) Respondents

During the survey period, a total of 744 valid questionnaires were collected. Females accounted for 54.57% of the respondents and males for 45.43%, reflecting a slightly higher participation rate among women. In terms of age distribution, 46.51% were aged 60–70, 32.93% were 71–80, and 20.56% were 81 or above. With regard to physical activity levels, 47.31% engaged in low-intensity activities, 35.22% in moderate-intensity activities, and 17.47% in high-intensity activities. Frailty status was distributed as follows: 45.83% were non-frail, 23.92% were pre-frail, and 30.24% were frail. The average

**Table 3. Microclimate data of measurement sites.**

| Place | Air temperature (°C) | | | Relative humidity (%) | | | Wind speed (m/s) | | | Black globe temperature (°C) | | |
|---|---|---|---|---|---|---|---|---|---|---|---|---|
| | Max. | Min. | Av. | Max. | Min. | Av. | Max. | Min. | Av. | Max. | Min. | Av. |
| ES | 15.1 | −2.3 | 9.5 | 87.8 | 20.2 | 41.2 | 0.6 | 0 | 0.2 | 15.4 | −2.5 | 5.9 |
| EP | 13.9 | −2.4 | 9.1 | 92.1 | 22.9 | 46.3 | 0.8 | 0 | 0.2 | 13.5 | −2.9 | 5.1 |
| WS | 14.2 | −2.2 | 9 | 95.3 | 25.5 | 50.4 | 3 | 0 | 0.8 | 13.8 | −3 | 5.2 |
| BS | 13.8 | −2.9 | 8.6 | 91 | 23.3 | 46.8 | 1 | 0 | 0.4 | 13 | −3.2 | 4.6 |
| RS1 | 14.5 | −2.5 | 9.1 | 90 | 20.6 | 43.5 | 0.6 | 0 | 0.1 | 13.7 | −2.1 | 5.2 |
| RS2 | 14.3 | −2.5 | 9.2 | 89 | 20.1 | 43.2 | 0.8 | 0 | 0.1 | 14.5 | −2 | 5.3 |

clothing thermal resistance was estimated at 1.57 clo for both the non-frail and pre-frail groups, and 1.79 clo for the frail group.

Fig 2 illustrates the distribution of older adults with different frailty levels across various activity spaces. In RS-type spaces, nearly half of the older adults were frail, suggesting a preference for resting on seating near roads and in areas closer to entrances and exits. In contrast, spaces such as EP and WS had a higher proportion of non-frail older adults, which may be attributed to the function of these spaces as rest areas within activity venues and their greater distance from entrances and exits, indicating that spatial function and proximity to access points influence venue choice.

(3) Thermal sensation vote

As shown in Fig 3, the distribution of TSV among older adults with different frailty levels varied in winter. Among non-frail subjects, the most frequent response was "cool" (TSV = −2). In contrast, "slightly cool" (TSV = −1) was the most common vote for those in the pre-frail and frail categories. Furthermore, a comparative analysis of the "neutral" (TSV = 0) responses reveals a positive correlation with frailty severity: the proportion of neutral votes increased from 11.44% in the non-frail group, to 20.22% in the pre-frail group, and further to 23.56% in the frail group.

(4) Thermal comfort vote

As shown in Fig 4, as the level of frailty increased, the proportion of older adults reporting a neutral TCV rose progressively to 6.74%, 9.55%, and 24.00%, respectively. Conversely, the proportion of subjects reporting feeling "comfortable" or "very comfortable" (TCV ≥ +1) gradually declined to 19.06%, 38.20%, and 52.00%, respectively. These trends indicate that under the same outdoor thermal condition, subjects with a greater degree of frailty tended to report higher thermal comfort.

Using TSV as the benchmark, the samples were grouped, and groups with fewer than 10 samples were excluded. The mean TCV was calculated for each group, and binary logistic regression analysis was performed to establish quantitative relationships between TSV and TCV across different frailty levels. The results are shown in Fig 5. The regression results showed good model fit for all three groups: non-frail ($R^2 = 0.67$, p = 0.188), pre-frail ($R^2 = 0.82$, p = 0.076), and frail ($R^2 = 0.87$, p < 0.049). For non-frail older adults, the most comfortable state occurred at TSV = 0.27, corresponding to TCV = 0.27. In the pre-frail group, the optimum shifted to TSV = 0.67 and TCV = 0.76, while in the frail group, it further increased to

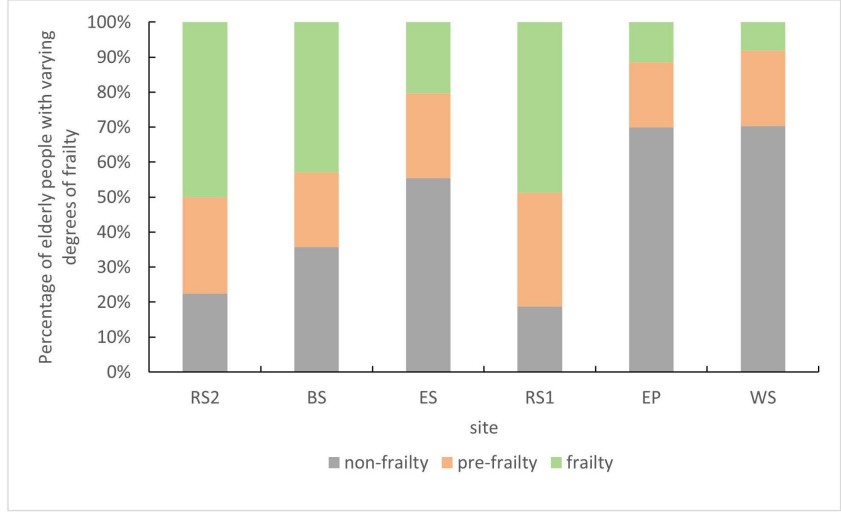

**Fig 2. Proportion of older people with different degrees of frailty at each measurement point.**

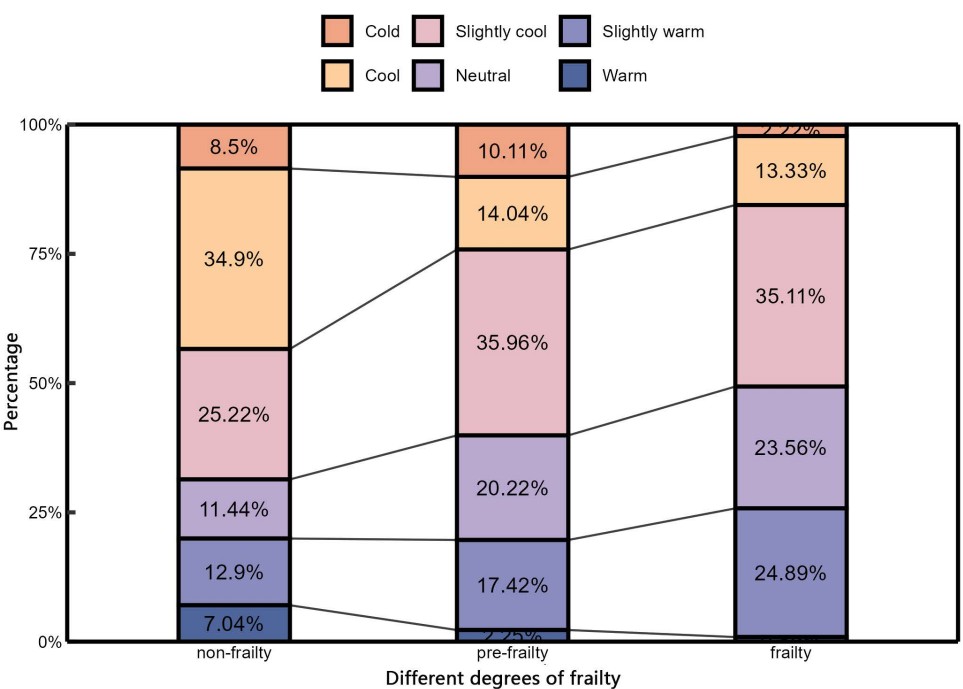

**Fig 3. TSV of older people with different degrees of frailty.**

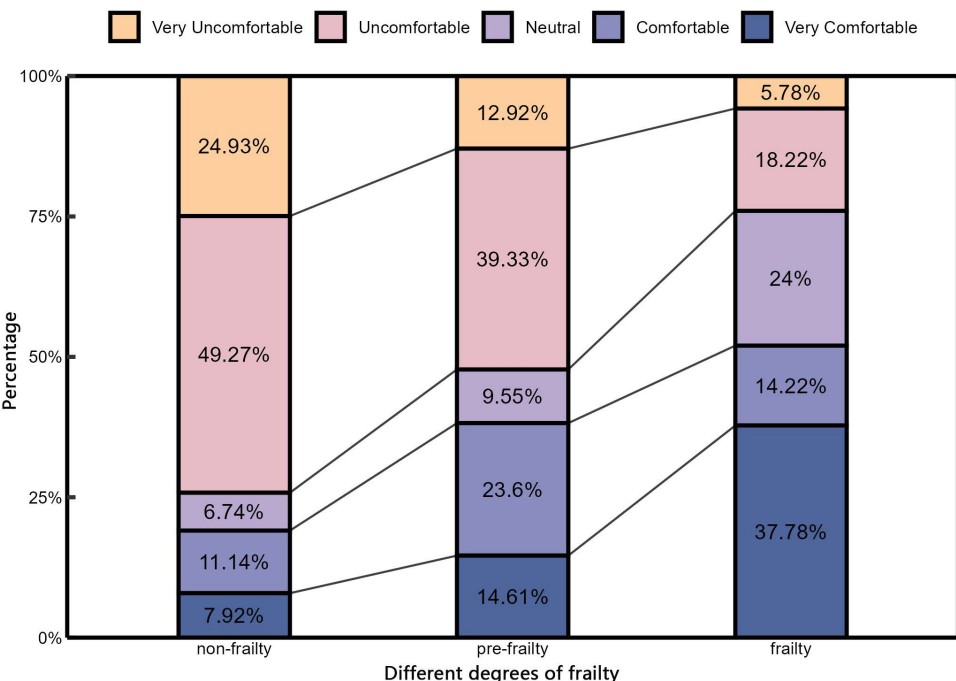

**Fig 4. TCV of older people with different frailty.**

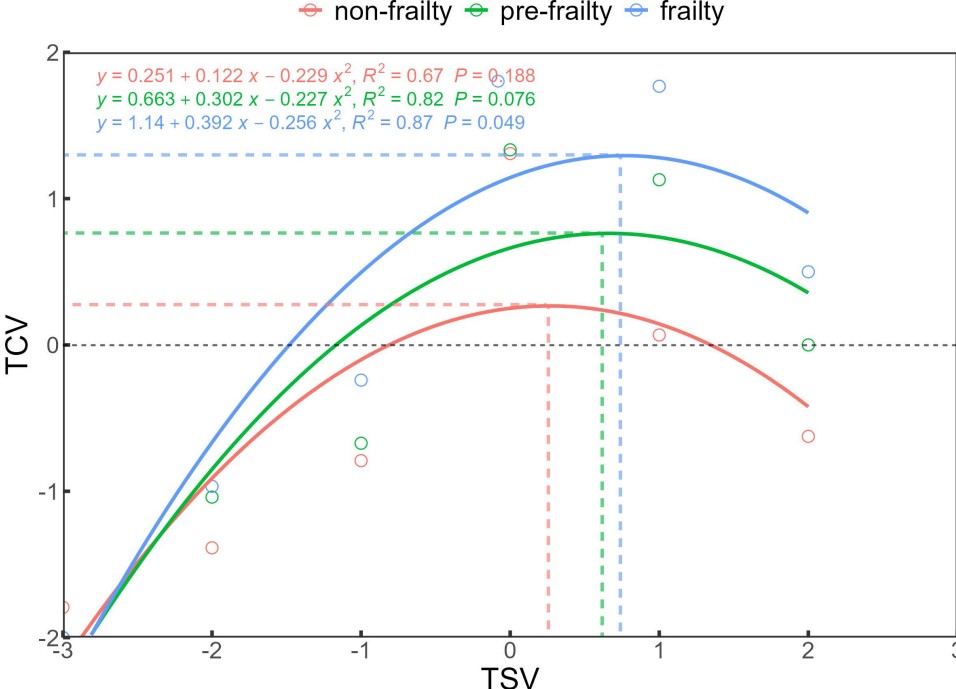

Fig 5. Correlation between TSV and TCV.

TSV = 0.77 and TCV = 1.29. These results indicate that as frailty severity increases, older adults achieve higher levels of comfort at higher TSV values. Under thermally neutral conditions (TSV = 0), frail older adults reported the highest thermal comfort. Furthermore, the TSV ranges corresponding to TCV > 0 were: −0.81 to 1.35 for the non-frail group, −1.17 to 2.50 for the pre-frail group, and −1.48 to 3.01 for the frail group. This suggests that in winter, the thermal comfort range widens with increasing frailty severity among older adults.

(5) Thermal environment preference

In terms of air temperature, most older adults preferred warmer conditions. As shown in Fig 6, this preference was most pronounced among the frail group, with 72.44% expressing a desire for higher temperatures. Regarding relative humidity, the differences among frailty levels were less distinct. A considerable proportion of participants across all groups preferred stable humidity conditions, while those who favored drier conditions outnumbered those preferring higher humidity. As for wind speed, the majority of respondents in all frailty categories favored maintaining current wind conditions, though a notable subset expressed a desire for reduced wind speed. With respect to solar radiation, the highest proportion of older adults across all frailty levels preferred increased solar exposure, reflecting a common desire for more sunlight in urban parks during winter in the region, which experiences hot summers and cold winters. Moreover, this preference strengthened with the severity of frailty, with the proportion rising from 57.77% (non-frailty) to 61.24% (pre-frailty) and 71.56% (frailty).

## 3.2 Differences in thermal benchmarks

### (1) Neutral PET and neutral PET range

To develop a winter thermal sensation model for older adults with varying frailty levels, PET values were grouped at 1°C intervals, and groups containing fewer than 10 samples were excluded. The MTSV was calculated for each PET

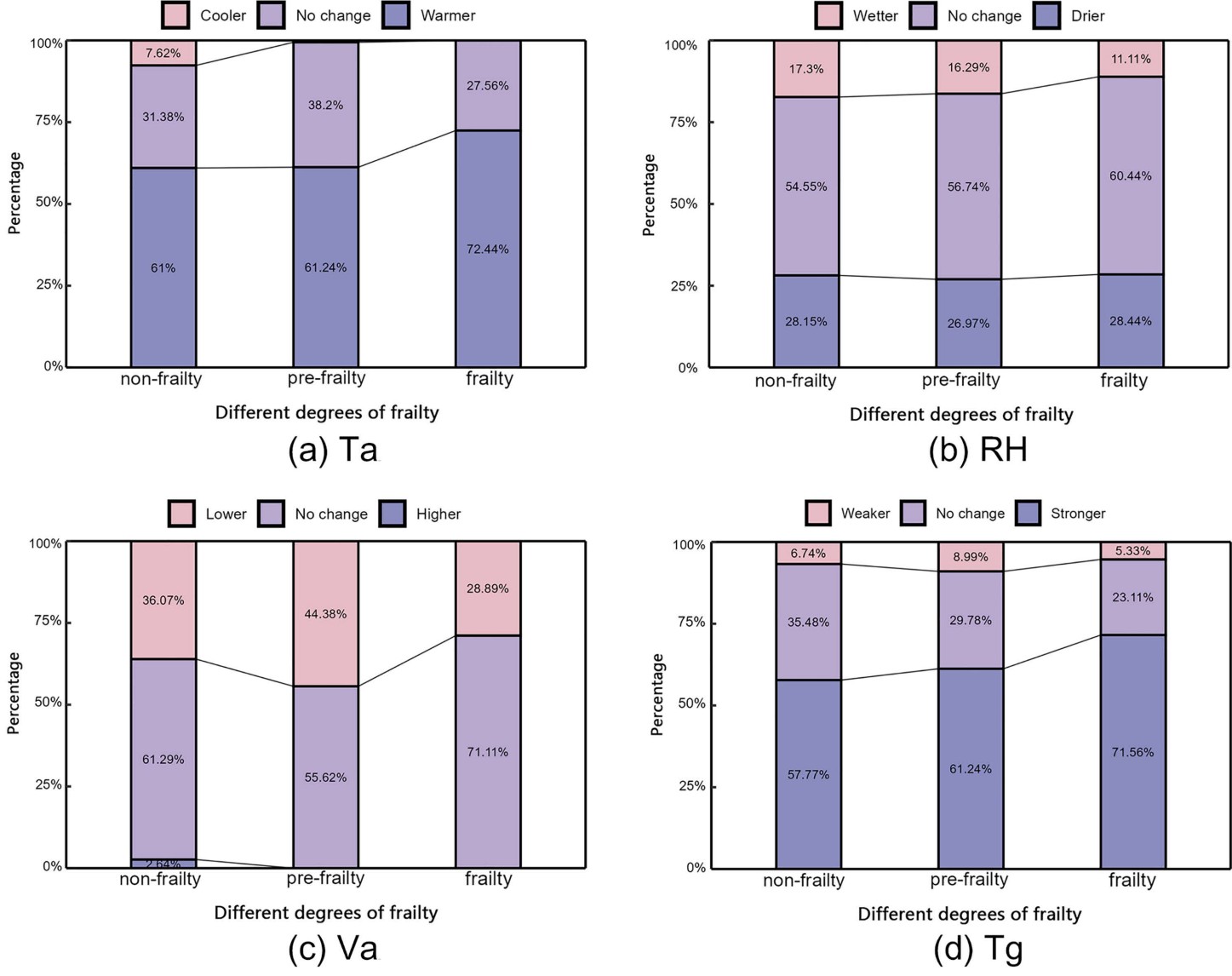

**Fig 6. Thermal environment preferences of older people with different degrees of frailty.**

interval, and a univariate linear regression model was fitted between PET (independent variable) and MTSV (dependent variable). The regression results showed strong linear relationships for all three groups: non-frail ($R^2 = 0.85$, $p < 0.001$), pre-frail ($R^2 = 0.87$, $p < 0.001$), and frail ($R^2 = 0.91$, $p < 0.001$). The regression equations are presented in Fig 7. The neutral temperature (NPET), defined as the PET value at which MTSV = 0, was derived by solving the regression equation for MTSV = 0, yielding NPET. The calculated NPET values were 10.29°C for the non-frail group, 9.60°C for the pre-frail group, and 8.65°C for the frail group, indicating that NPET decreases as frailty severity increases. The neutral temperature range (NPETR), defined as the PET interval in which MTSV falls between −0.5 and +0.5 [55], was 7.86–12.72°C for the non-frailty, 7.21–12.18°C for the pre-frailty, and 5.95–11.35°C for the frailty. Compared to the severely frailty group, the non-frailty group exhibited a higher lower and upper NPETR limit, suggesting that greater frailty is associated with broader thermal tolerance. The slope of the regression line reflects the sensitivity of thermal sensation to changes in PET.

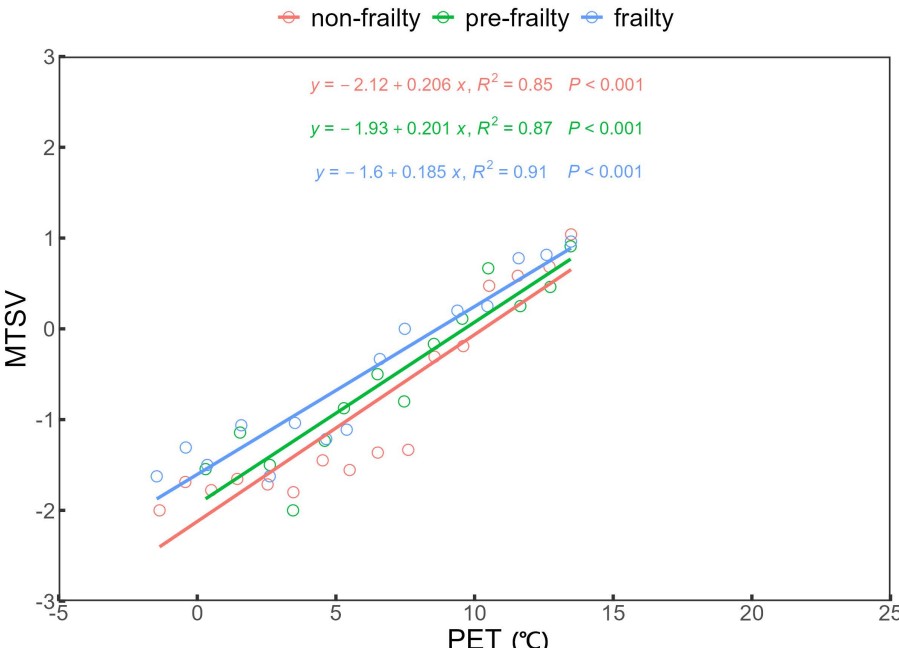

**Fig 7. Correlation between PET and MTSV.**

The slopes were 0.206 for the non-frailty, 0.201 for the pre-frailty, and 0.185 for the frailty, indicating a gradual decline in thermal sensitivity with increasing frailty.

### (2) Thermal Acceptability Voting (TAR)

The thermal unacceptability rate refers to the percentage of votes indicating thermally unacceptable conditions. According to ASHRAE Standard 55, the thermal acceptability range (TAR) is defined as the temperature range considered acceptable by 90% of occupants under strict conditions [56], meaning the unacceptability rate does not exceed 10%. Following this standard, PET values for older adults with different frailty levels were grouped at 1°C intervals. The unacceptability rate (proportion of votes indicating 'unacceptable') was calculated for each interval, and a binary logistic regression model was fitted with PET as the independent variable and thermal acceptability as the dependent variable. The results are shown in Fig 8. The regression results showed good model fit for all three groups: non-frail ($R^2 = 0.94$, $p < 0.001$), pre-frail ($R^2 = 0.92$, $p < 0.001$), and frail ($R^2 = 0.94$, $p < 0.001$). The estimated winter TAR for older adults was 6.04–11.19°C for the non-frailty group, 5.67–12.01°C for the pre-frailty group, and 4.68–13.07°C for the frailty group. These results further indicate that thermal tolerance increases with the severity of frailty, as evidenced by the broader TAR in more frail individuals.

### (3) Optimal temperature

Probit regression analysis was employed to determine the optimal temperature for older adults with varying degrees of frailty. In this analysis, the probability of selecting "no change" was randomly allocated to the "warm" and "cold" response categories. The preferred PET was identified as the intersection point of the two resulting logistic regression curves. As shown in Fig 9, the probit regression curves intersect at PET values of 11.9°C for the non-frail group and 16.2°C for the pre-frail group. For the frail group, the intersection occurred at 29.0°C. Importantly, this value lies far outside the range of PET values actually recorded during the winter survey. Therefore, the 29.0°C estimate is a statistical extrapolation that lacks empirical support and should not be interpreted as a realistic comfortable temperature in winter. Instead, the key

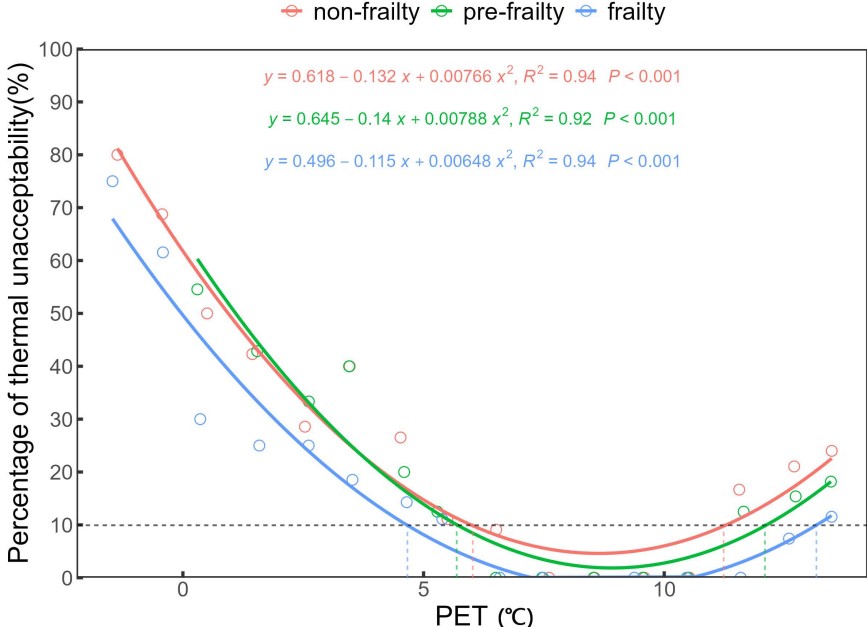

**Fig 8. Correlation between PET and TAR.**

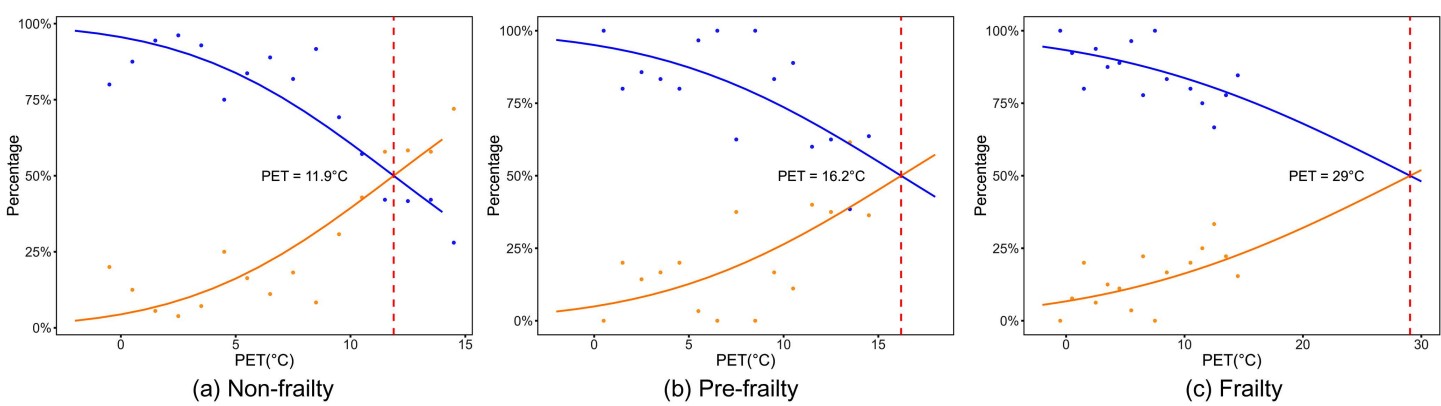

**Fig 9. Preferred temperature for older people with different degrees of frailty.**

finding from the probit analysis is the consistent ordering: frail > pre-frail > non-frail, indicating that frail older adults have a stronger relative preference for warmer conditions. Future studies with all-season data or controlled thermal exposures are needed to reliably quantify the absolute preferred temperature for frail individuals. These results demonstrate that thermal preference increases with frailty severity, with significantly frail older adults exhibiting a substantially higher preferred temperature than the other two groups.

**(4) Optimal time period**

Taking a typical winter day (December 21, 2024) as examples, meteorological data were collected hourly at each measurement point, and the corresponding PET values were calculated based on a standard human model, illustrating the hourly variation in PET from 9:00 AM to 18:00 PM [33]. The visualization results are presented in Fig 10. Based on the previously analyzed neutral PET range, variations in PET values were observed across different measurement locations. In summer-hot and winter-cold regions, during winter, the optimal outdoor periods for non-frail and pre-frail older adults are as follows: at EP, WS, and RS1 from 12:00–14:00; at ES from 11:00–14:00; at RS2 from 12:00–16:00; and at BS from 12:00–14:00 and 15:00–16:00. For severely frail older adults, in addition to the above periods, suitable times extend to 16:00–17:00 at ES, 14:00–15:00 at BS, and 10:00–11:00 at EP. The results indicate that the time windows during which older people experience thermal neutrality in urban parks of Nanyang City are concentrated around and after lunchtime. For most of the day, older adults are exposed to varying levels of cold stress. Therefore, it is evident that on typical winter days, urban parks remain unsuitable for older people's activities during most hours. There is an urgent need to enhance thermal comfort and increase the utilization efficiency of urban parks in winter.

## 3.3 Factors affecting thermal sensation

A parallel line test was first conducted to assess the model assumptions. The survey data across the three frailty levels of older adults all satisfied these assumptions, making them suitable for subsequent ordered regression analysis. However, the ordered regression results indicated that only a few variables, such as temperature and humidity, were statistically significant ($p < 0.05$). This limited outcome was likely due to the relatively small sample size within each frailty subgroup and the multicollinearity among predictors, which reduced the statistical power of the multivariate model to detect the individual contributions of personal, social, and psychological factors.

　　Given these limitations, we employed univariate regression analysis using Spearman correlation as a complementary approach to explore the relationships between each independent variable and thermal perception outcomes (TSV, TCV, and TAV). Spearman correlation was selected for three reasons. First, it does not assume linearity or normality, making it robust for analyzing ordinal data such as Likert-scale thermal sensation votes. Second, it allows for the examination of individual factor contributions without the risk of overfitting due to multicollinearity. Third, it is widely used in exploratory outdoor thermal comfort studies to identify key influencing factors prior to more complex modeling. This approach enabled

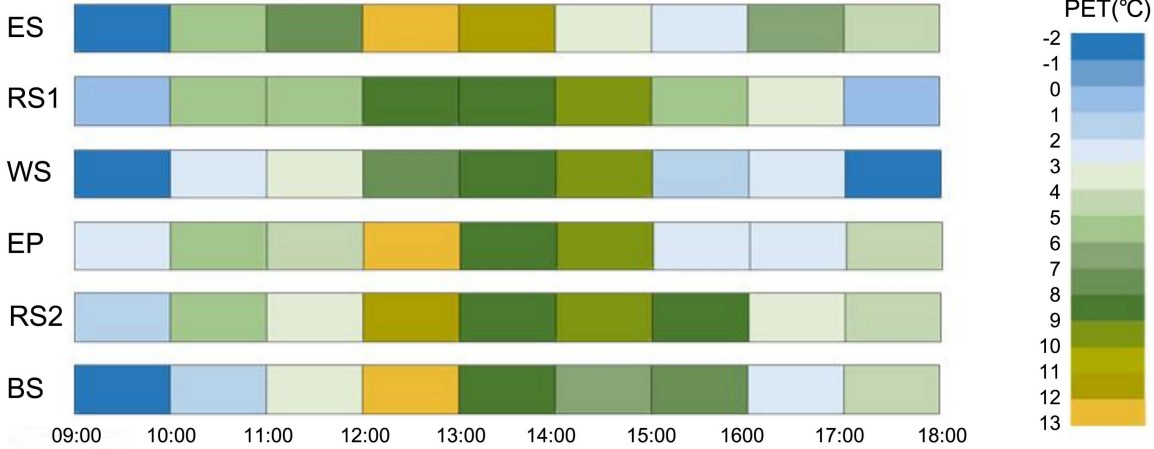

**Fig 10. Optimal time period for older people with different frailty.**

us to reveal how the influence of personal, social, and psychological factors varies across frailty levels..The results of this analysis are presented in Fig 11.

As shown in Fig 11, the influencing factors have the strongest association with TSV, followed by TCV, and then TAV. While commonalities exist in the factors influencing outdoor thermal sensation among older adults with varying frailty levels, distinct differences are also observed.

The ranking of factors affecting thermal perception by frailty level is as follows:

Non-frailty: Ta > Tg > satisfaction > activity > gender > companionship > age > frequency;

Pre-frailty: Ta > Tg > activity > age > gender > companionship > satisfaction > frequency > monthly income;

Frailty: Ta > Tg > companionship > age > activity > satisfaction > gender > Va > purpose.

Overall, physical environmental factors—particularly air temperature (Ta) and globe temperature (Tg)—exert the strongest influence on thermal perception across all frailty levels. Psychological factors such as satisfaction play a notable role among non-frail older adults. Among pre-frail individuals, personal factors like age and activity level show greater impact, while for frail older adults, social factors such as companionship significantly shape thermal perception.

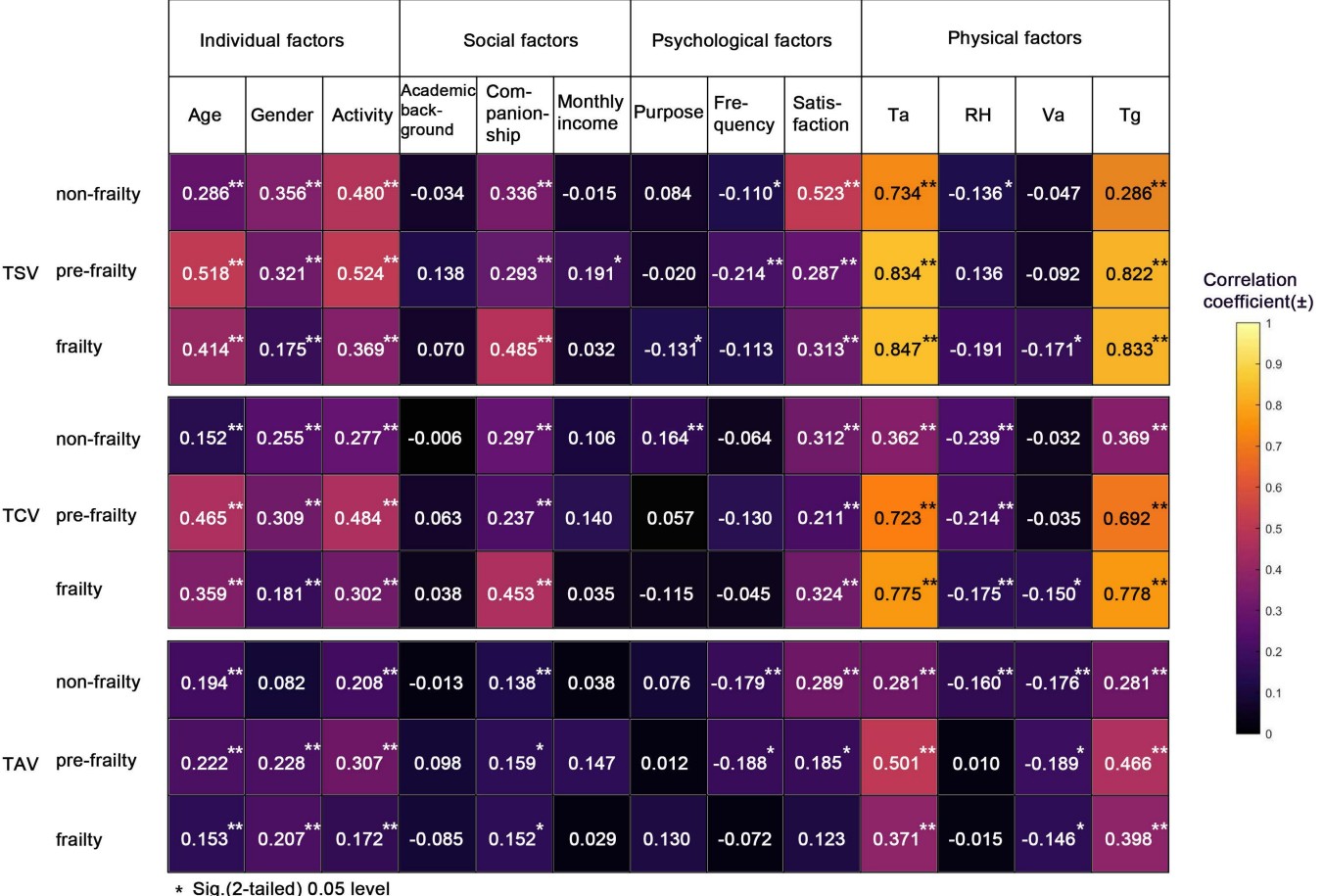

**Fig 11. The correlation between each variable and thermal sensation.**

**(1) Personal factors**

The three selected personal factors—age, gender, and activity intensity—all showed significant positive correlations with TSV among older adults across the three frailty levels. For non-frail and pre-frail individuals, activity intensity had the most pronounced effect on TSV, whereas for those with marked frailty, age was the most influential factor, followed by activity intensity. This suggests that activity intensity exerts a substantial influence across all frailty levels, with thermal sensation increasing as activity intensity rises. Additionally, women generally reported stronger thermal sensations than men, although the effect of gender diminished with increasing frailty severity.

In terms of TCV, age had the greatest impact among pre-frail older adults, followed by those with marked frailty, and was least influential in the non-frail group. Regarding TAV, gender did not show a significant effect in non-frail older adults.

The effect of personal factors on TCV and TAV of the three levels of frailty was similar to that of TSV, but the degree of influence was less than that of TSV.

**(2) Social factors**

Social companionship demonstrates a significant positive correlation with TSV, TCV, and TAV across all three frailty levels among older adults, indicating that the presence of companions leads to higher thermal perception compared to being alone. Monthly income is only associated with TSV in the pre-frail stage, with no significant correlations observed at other frailty levels. Educational background shows no statistically significant relationship with TSV, TCV, or TAV in any of the three frailty groups.

The effect of companionship on TSV and TCV is most pronounced among older adults with marked frailty, suggesting that social accompaniment can more effectively improve their outdoor thermal comfort in winter compared to those without companionship. However, its influence on TAV does not differ significantly across the three frailty levels.

Overall, while the impact of social factors on older adults is generally modest, companionship exerts a relatively stronger influence—particularly among those with pronounced frailty.

**(3) Psychological factors**

Regarding the three selected psychological factors—purpose, frequency, and satisfaction—satisfaction demonstrated a significant positive correlation with TSV, TCV, and TAV across all three frailty levels, except for TAV among the significantly frail group. This indicates that higher satisfaction levels generally contribute to improved thermal perception and comfort. Satisfaction exerted the strongest influence on TSV in non-frail older adults, underscoring its important role in shaping outdoor thermal perception in this subgroup.

From the perspective of thermal acceptance (TAV), the impact of satisfaction diminished as the degree of frailty increased. Frequency, on the other hand, showed no significant correlation with TSV, TCV, or TAV among non-frail older adults.

Overall, these results suggest that among psychological factors, satisfaction is the most consistent and influential predictor of thermal perception, while other factors such as purpose and frequency exhibit limited effects.

**(4) Physical factors**

The most influential factors affecting TSV are Ta, followed by Tg. As the degree of frailty increases, the impact of both Ta and Tg on TSV becomes more pronounced.RH shows a significant negative correlation with TSV only in non-frail older adults, with no significant association observed in the other two frailty groups. Similarly, Va exhibits a significant negative correlation solely among the severely frail group, though with a relatively small correlation coefficient, and shows no significant relationship with TSV in the other two groups.

Regarding TCV, Ta, Tg, and RH were significantly correlated among non-frail and pre-frail older adults, with RH showing a negative association. For those with marked frailty, all four physical factors demonstrated significant correlations with

TCV. Among them, Ta and Tg remained the most influential, while both RH and Va were negatively correlated with TCV, indicating that higher humidity and wind speed correspond to lower comfort levels.

In terms of TAV, Ta and Tg showed positive correlations to varying degrees across frailty levels, whereas Va was consistently negatively correlated. RH was significantly and negatively correlated with TAV only in the non-frail group and showed no significant association in the other two groups.

### 3.4 Differences in thermal adaptation behavior

To investigate differences in outdoor thermal adaptation behaviors among older adults with varying levels of frailty during winter, our research team conducted follow-up observations of respondents at six measurement sites. In response to uncomfortable outdoor conditions, older adults in urban parks exhibited the following adaptive behaviors: leaving the area, seeking sunlight, increasing activity intensity, adding clothing, and drinking hot water. The distribution of these behaviors across frailty levels and measurement sites is shown in Fig 12.

Distinct differences in thermal adaptation behaviors were observed across sites, indicating that the behaviors of older adults are significantly influenced by their immediate environment. Older individuals tended to adapt to thermal discomfort by utilizing available environmental features. For instance, at the EP site near the sports field, individuals often used fitness equipment to increase activity intensity, whereas at the BS site adjacent to buildings, people were more likely to leave the area and move indoors. At the ES site, located near park entrances and exits, respondents showed a greater tendency to leave the park entirely and return home. As this site also functioned as a dance area, some older adults adapted to unfavorable weather by dancing—effectively increasing their activity intensity.

At the RS1, RS2, and WS sites, which feature seating near roads or water bodies, older adults more frequently sought sunny spots or walked along paths to raise their activity levels. In contrast, the proportions of adding clothing and drinking hot water were relatively low across all sites.

Notably, at both RS sites, as frailty levels increased, more older adults chose to leave, while fewer opted to increase activity levels. This suggests that in the face of adverse winter conditions, frailer individuals—likely due to declining physical capacity—tend to avoid active adaptation and instead prefer to exit the uncomfortable environment. A similar pattern was observed at the ES and BS sites. Furthermore, across all sites except WS, frail older adults were less inclined to raise activity levels and more likely to seek sunlight as a means of adaptation.

## 4. Discussion

### 4.1 Differences in thermal benchmarks

Comparing to the studies undertaken in similar summer-hot and winter-cold regions, the NPET of the older people in this study was smaller than that of the mixed population in Chengdu, Changsha and Hangzhou [25,57], and higher than that of older people study in a community park in Huangshan, China [35]. Therefore, based on current research, the NPET of older people in hot summer and cold winter regions is less than that of mixed populations or younger individuals, but more studies are needed to confirm this conclusion. In this study, older people with varying degrees of frailty all require a slightly warmer thermal sensation to achieve optimal thermal comfort. Moreover, frail older people need a higher thermal sensation, indicating that they have slower metabolism and produce less heat internally. In the cold winter, they need more external thermal sensation to reach a comfortable state, which is consistent with the findings from Huangshan and Chengdu studies [35]. However, compared to the study in Huangshan, both the upper and lower limits of NPET and NPETR in this study were nearly 6°C higher. This temperature difference is relatively high, and it may be related to the latitude and longitude of the study area, or it could be associated with the micro-environment around the measurement site. Therefore, more research is needed to enrich the OTC for older people in hot-summer and cold-winter regions, to identify regular causes.

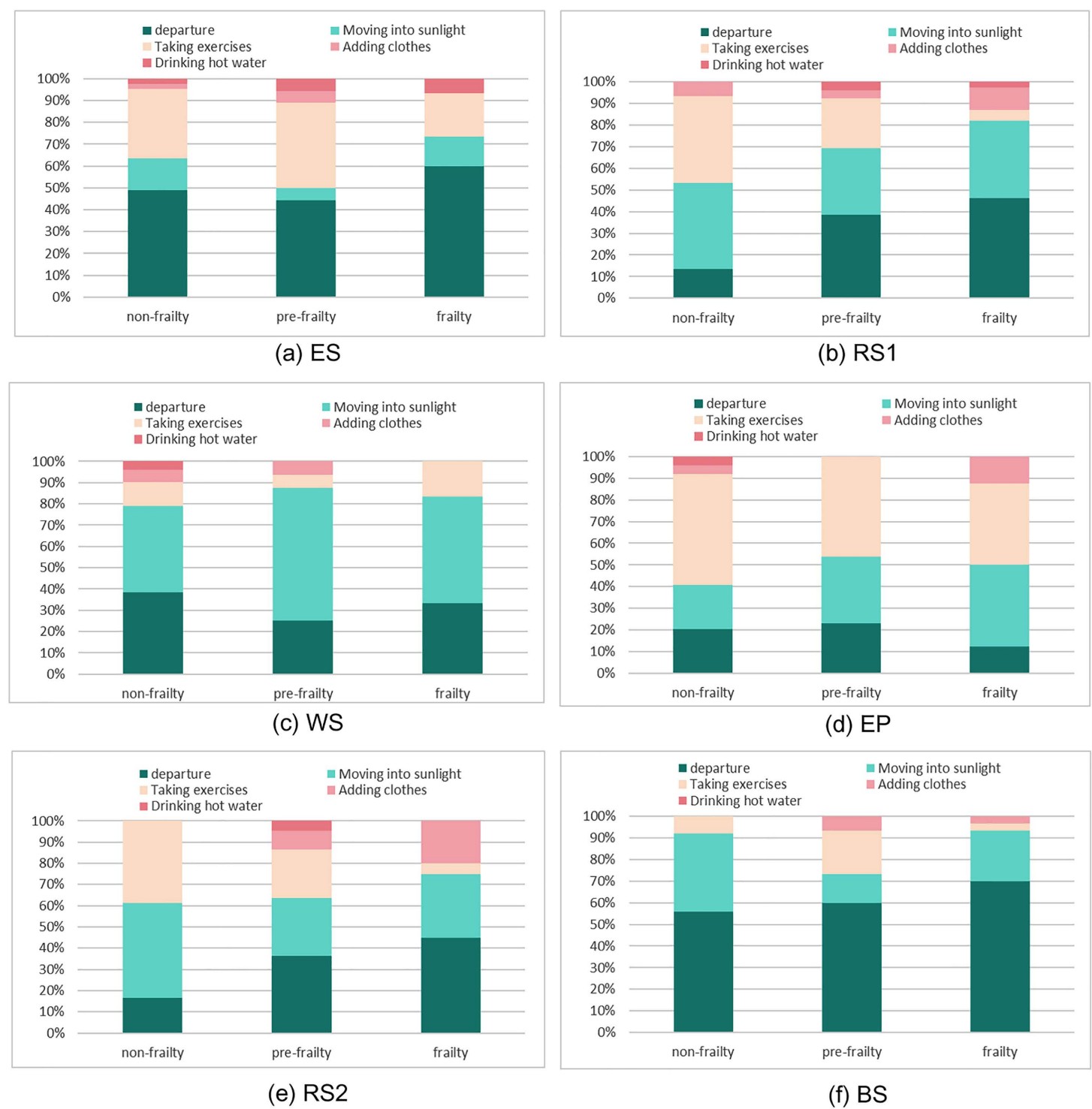

**Fig 12. Adaptation behavior of older people with different degrees of frailty at each measurement point.**

The probit regression analysis (Section 3.2) yielded an estimated preferred PET of 29.0°C for frail older adults. However, this value is derived by extrapolating the logistic curves far beyond the observed PET range. Statistically, such extrapolation assumes that the relationship between PET and choice probability remains logistic outside the data range, an assumption that cannot be verified with the current dataset. Therefore, this 29.0°C figure should not be cited or used as a quantitative benchmark. Its only scientifically defensible use is to indicate a qualitative ordering: among the three frailty groups, frail individuals expressed the strongest desire for warmer conditions, followed by pre-frail, and then non-frail. This ordering is consistent with the neutral temperature results (NPET decreasing with frailty) and with their reduced metabolic heat production. To reliably quantify the preferred temperature of frail older adults, future research should either (a) conduct measurements across all seasons (including spring and autumn) to capture a wider PET range, or (b) use controlled thermal chamber experiments with frail participants, which can safely expose them to a broader temperature spectrum without relying on statistical extrapolation. Nevertheless, this finding carries meaningful implications. The substantially higher preferred PET among frail older adults, compared with the non-frail (11.9°C) and pre-frail (16.2°C) groups, suggests that frail individuals have a stronger psychological and physiological preference for warmer environments when exposed to cold winter conditions. This aligns with their reduced metabolic heat production and impaired thermoregulatory capacity, which necessitate greater external heat input to achieve thermal comfort. In practical terms, this indicates that for frail older adults, winter outdoor spaces should prioritize maximizing solar radiation exposure, providing radiant heating elements (e.g., heated seating), and offering enclosed or semi-enclosed warm shelters within urban parks. Future studies should aim to validate this finding by conducting all-season measurements that include transitional seasons (spring and autumn) or by employing controlled thermal chamber experiments with frail participants to obtain preference data across a broader PET range without relying on model extrapolation.

### 4.2 Comparison of factors affecting thermal sensation

#### (1) Physical factors

Among the physical factors, Ta has the greatest influence on outdoor thermal sensation, followed by Tg, and its influence is greater than Va, which is similar to Changsha [25], Harbin [58,59]. The results are the same as those in other parts of China, so it can be concluded that for different populations in China and even different parts of the world, Ta is the most important physical parameter affecting outdoor thermal comfort, followed by Tg [60–62]. The higher the degree of frailty, the greater the influence of Ta, indicating that with the decline of human function, the weak older people are more sensitive on Ta. However, in low temperatures around the world, the influence of Va is greater than Tg in a few regions, because convective heat loss becomes more important at extreme low temperatures [63,64]. RH is usually considered to be the least important physical factor, however, its effect on pre-frail older people was greater than that of Va in this study. The results may be influenced by the specific environment at the time of measurement, because when the temperature is higher, the humidity in the air increases, and so does the effect of RH [65].

#### (2) Personal factors

There are many studies on outdoor thermal comfort by age group, and most of the research results show that the older people are less sensitive to outdoor thermal sensation than the young [59,66]. It is also believed that the older the older people are, the smaller the NPET and the wider the NPETR, and the less sensitive to the outdoor thermal environment [35]. This study shows that the greater the degree of frailty, the smaller the NPET and the wider the NPETR, making it less sensitive. This finding is consistent with the results of age stratification. However, it doesnot mean that age-based classification can replace the assessment of frailty, rather it suggests that older people with high frailty and advanced age are the least sensitive to outdoor conditions in winter and should be given special attention. Generally, it is believed that gender has a very small impact on OTC, which can be disregarded [67,68]. However, in this study, gender had a significant

impact on outdoor thermal sensation for older people with varying degrees of frailty. As the degree of frailty increased, the influence of gender gradually diminished. The intensity of activity had a greater effect on OTC for older people with different degrees of frailty. This is because increased activity intensity generates more heat, making them less sensitive to the cold winter climate and having a lower NPET [26,69].

However, it should be noted that a study of older adults in the Hong Kong, China came to the opposite conclusion, that individual factors had no effect on OTC [38]. Therefore, there is no consensus for the influence of individual factors on OTC, which is affected by multiple local social and cultural factors and living habits, and needs to be analyzed according to the specific situation of the study area.

(3) **Social and psychological factors**

Social and psychological factors are considered to be indirect factors that have less effect on OTC [20]. Companions had a significant effect on the older people with different degrees of frailty, which was consistent with the results of Shooshtarian S et al. [70]. Although the study involved a mixed population, the impact of companionship on TSV and TCV in significantly frail older people was more pronounced compared to other groups of older people. This indicates that frail older people require more companionship, and those with companions have a warmer thermal sensation. This fully demonstrates the influence of social factors on outdoor thermal sensation in different frail older people. There has been limited research on the impact of overall environmental satisfaction on OTC. In fact, satisfaction is an important influencing factor. This study shows that as overall satisfaction increases, the TSV and TCV of older people with varying degrees of frailty also increase. Therefore, creating a satisfactory park environment has a significant promoting effect on the OTC of older people.

### 4.3 Comparison with indoor and outdoor studies of older people with different degrees of frailty

This study found that as the degree of frailty increases, the outdoor thermal sensitivity of older people gradually decreases. This reduced sensitivity can be attributed to physiological and behavioral mechanisms associated with frailty. Physiologically, frailty involves age-related declines in autonomic nervous system function, including impaired vasoconstriction and reduced sweating capacity, which blunt the body's ability to detect and respond to thermal stimuli [48,71]. Behaviorally, frail older adults may have reduced activity levels and spend less time outdoors, potentially leading to diminished thermal perception through reduced environmental exposure and adaptation [72].

Our finding aligns with studies on older adults with varying frailty levels in China's hot-humid regions during summer [48], which also reported blunted thermal perception in frail individuals. However, it contrasts with a study on extreme high-temperature weather outdoors in Beijing [73], which found that chronic disease patients had higher neutral temperatures than healthy individuals. This discrepancy may be explained by differences in health status definitions—chronic disease does not necessarily equate to frailty, as frailty encompasses multidimensional physiological decline beyond specific disease diagnoses [74]. Additionally, the Beijing study used UTCI rather than PET, and these indices incorporate different thermophysiological assumptions, which may affect sensitivity estimates.

Interestingly, an indoor study of frail older adults in China's hot summer and cold winter region found that they were more sensitive to heat and had a narrower comfortable temperature range [49]—the opposite of our outdoor findings. This divergence highlights the critical role of environmental context: indoors, frail individuals are exposed to stable, controlled conditions where their impaired thermoregulatory systems may struggle to maintain comfort; outdoors, they face dynamic environmental stimuli that may exceed their perceptual thresholds, resulting in blunted sensation. Furthermore, frail older adults may be less capable of taking timely adaptive actions (e.g., seeking shade, adjusting clothing) when outdoors, making them more vulnerable to extreme temperatures despite reporting lower sensitivity [48].

The above differences may also stem from methodological factors, including different frailty assessment tools (this study used the FRAIL scale; Zhou et al. [49–51] used the Fried Frailty Phenotype), climate backgrounds, cultural

differences in clothing insulation, and questionnaire design. Future research should systematically compare the performance of UTCI and PET in assessing outdoor thermal comfort for older adults with different frailty levels, to identify the most appropriate evaluation indicators for regions with hot summers and cold winters.

## 4.4 Design strategy

From the above research, it can be seen that there are obvious differences in OTC and heat adaptation behaviors of older people with different degrees of frailty. Therefore, designers should fully consider this point and carry out refined design of urban parks.

The location for public spaces in urban parks should fully consider accessibility for frail older people. The site should be close to entrances and exits and near internal roads, making it more friendly for them. In the deeper areas of urban parks, public spaces suitable for non-frail older people should be set up, ideally accompanied by activity equipment or water bodies.

Ta is the most significant factor affecting outdoor thermal comfort. Different winter outdoor temperatures should be created through plant configuration and building structures. Ground surfaces with varying degrees of paving should be set up in areas where older people gather, increasing the ground radiation temperature to suit the needs of older people with different levels of frailty. Additionally, heated seats can be installed, replacing stone seats with warmer wooden ones. Evergreen trees should be planted along the prevailing winter winds to block the cold air.

Although the intensity of activities has a strong impact on OTC, the proportion of older people with weakened physical conditions seeking sunlight is higher than those who engage in enhanced activities. Therefore, it is essential to first increase the number of sunlit seats for use by the older people, especially in areas where frail older people gather. Activities are beneficial for both their physical and mental health. Thus, setting up interesting activity equipment to spark the interest of frail older people is advantageous.

Similarly, installing activity equipment in areas with gathered non-frail older people is also very necessary. Set up chess, ballroom dancing, and theater venues in appropriate locations within the park to cater to the needs of older people of different age groups, thereby increasing their social opportunities. Since companionship is a crucial factor influencing OTC, enriching the spiritual lives of older people and enhancing their sense of social participation should be prioritized. It is essential to conduct research on the psychological needs of older people and tailor the parks functions, plant configurations, architectural styles, and hard paving according to their specific requirements. This will improve overall satisfaction with urban parks among older people and have a positive impact on OTC.

## 4.5 Limitations of this study

This study uses different levels of frailty as the criterion for categorizing older people, selecting a typical urban park in Nanyang City, a typical city in hot-summer and cold-winter in China, as the research subject. It employs a combination of subjective questionnaires and objective measurements to investigate differences in OTC among older people with varying degrees of frailty. Within the limited scope, it explores the intersection of physiology and architecture, providing a reference for OTC studies in similar regions. However, there are still certain limitations:

(1) Only the OTC of older people with different degrees of frailty in winter was investigated. Subsequent studies will supplement the OTC study in summer to study the changes of OTC under extreme cold and heat weather.

(2) Due to the relatively small number of older people in urban parks in cold weather in winter, such as WS measurement points, there are certain errors and uncertainties in the percentage of older people with different degrees of frailty facing environmental changes in different locations.

(3) Although the article discusses the influence of physical, personal, social and psychological factors on OTC, OTC is affected by a variety of factors, and the article lacks consideration of many other factors like hot expectation, hot history, culture and SVF.

(4) The article only takes one park as a case study, and its scientificity and extensibility are open to debate. In future research, different city parks in different latitudes and longitudes in hot summer and cold winter areas should be investigated and compared to draw relatively reliable conclusions.

(5) The probit-derived preferred PET for the frail group (29.0°C) fell outside the empirical PET range. This value is a statistical extrapolation and should not be interpreted as a valid quantitative estimate. Consequently, we treat it only as a directional indicator (frail > pre-frail > non-frail). Future studies should employ all-season measurements or controlled chamber experiments to obtain reliable absolute preferred temperatures for frail older adults.

## 5. Conclusion

This paper combines objective field measurement and subjective questionnaire survey, and uses FRAIL scale as the classification basis to explore the differences in thermal comfort of older people with different degrees of frailty, and determine the OTC thermal baseline of older people in urban parks in hot summer and cold winter areas. The main findings of this study are as follows:

(1) Frail older people tended to choose the space near the road and close to the entrance and exit, while non-frail older people preferred the space near sports equipment and water.

(2) In regions with hot summers and cold winters, in outdoors older people need a slightly warmer thermal sensation to achieve optimal thermal comfort. The higher the degree of frailty, the higher the level at which they reach this most comfortable state, and the wider the range of thermal comfort.

(3) The NPET for non-frailty is 10.29°C, for pre-frailty is 9.60°C, and for frailty is 8.65°C. The higher the degree of frailty, the lower the neutral temperature. The NPETR for non-frailty is 8.66–12.72°C, for pre-frailty is 7.21–12.18°C, and for frailty is 5.95–11.35°C. The slope of the correlation line between PET and MTSV indicates that as the degree of frailty increases, the thermal sensitivity of older people gradually decreases.

(4) Frail older adults expressed a stronger preference for warmer conditions compared to pre-frail and non-frail groups, although the absolute preferred PET could not be reliably quantified within the winter measurement range. Each measurement point has different optimal entry time, and the most comfortable time to visit the park is 12:00–14:00 in winter.

(5) Physical factors including temperature and solar radiation, personal factors including age, gender and activity intensity, and social psychological factors including companionship and satisfaction have a significant impact on OTC of the older people, but the intensity of the impact varies with different degrees of frailty.

(6) The adaptive behavior of the older people with different degrees of frailty is greatly influenced by the specific environment, and the older people tend to rely on the surrounding environment to choose thermal adaptive behavior. Facing undesirable cold weather, the frail older people tend to leave or seek sunshine while the non-frail older people tend to increase physical activities.

## Supporting information

**S1 Table. Proportion of older people with different degrees of frailty at each measurement point.**
(XLSX)

**S2 Table. Field research data summary.**
(XLSX)

## Acknowledgments

The authors would like to express appreciation to all the survey participants in the park for their support and cooperation in this research.

## Author contributions

**Conceptualization:** Dong Yan, Biao Wang.

**Data curation:** Dong Yan, Ran Chen.

**Formal analysis:** Dong Yan, Ran Chen.

**Funding acquisition:** Dong Yan, Zhiyuan Liu.

**Investigation:** Dong Yan.

**Methodology:** Dong Yan, Biao Wang.

**Project administration:** Zhiyuan Liu.

**Resources:** Dong Yan, Peng Zhang, Biao Wang.

**Software:** Dong Yan, Peng Zhang, Ran Chen.

**Supervision:** Zhiyuan Liu.

**Validation:** Biao Wang.

**Visualization:** Dong Yan, Peng Zhang.

**Writing – original draft:** Dong Yan.

**Writing – review & editing:** Dong Yan, Biao Wang.

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
