## [Decision Letter · Decision Letter 0]

26 Feb 2026

PONE-D-25-66927Winter outdoor thermal comfort of older people with different degrees of frailty: a study in Nanyang city, ChinaPLOS One

Dear Dr. Wang,

Thank you for submitting your manuscript to PLOS ONE. After careful consideration, we feel that it has merit but does not fully meet PLOS ONE’s publication criteria as it currently stands. Therefore, we invite you to submit a revised version of the manuscript that addresses the points raised during the review process.

We look forward to receiving your revised manuscript.

Kind regards,

Xuan Ma

Academic Editor

PLOS One

Journal Requirements:

[This research was funded by the Science and technology project of Henan Province (252102320315). Housing and Urban Rural Construction Science and Technology Plan Project (HNJS-2024-R1), and Henan Province soft science project (242400410262, 252400410248).].

3. Thank you for stating the following in your manuscript:

[This research was funded by the Science and technology project of Henan Province (252102320315). Housing and Urban Rural Construction Science and Technology Plan Project (HNJS-2024-R1), and Henan Province soft science project (242400410262, 252400410248).]

[This research was funded by the Science and technology project of Henan Province (252102320315). Housing and Urban Rural Construction Science and Technology Plan Project (HNJS-2024-R1), and Henan Province soft science project (242400410262, 252400410248).]

6. We note that Figure 1 in your submission contains images which may be copyrighted. All PLOS content is published under the Creative Commons Attribution License (CC BY 4.0), which means that the manuscript, images, and Supporting Information files will be freely available online, and any third party is permitted to access, download, copy, distribute, and use these materials in any way, even commercially, with proper attribution. For more information, see our copyright guidelines: http://journals.plos.org/plosone/s/licenses-and-copyright.

7. We note that Figure 1 in your submission contains map and satellite images which may be copyrighted. All PLOS content is published under the Creative Commons Attribution License (CC BY 4.0), which means that the manuscript, images, and Supporting Information files will be freely available online, and any third party is permitted to access, download, copy, distribute, and use these materials in any way, even commercially, with proper attribution. For these reasons, we cannot publish previously copyrighted maps or satellite images created using proprietary data, such as Google software (Google Maps, Street View, and Earth). For more information, see our copyright guidelines: http://journals.plos.org/plosone/s/licenses-and-copyright.

8. Please ensure that you refer to Figure 6 in your text as, if accepted, production will need this reference to link the reader to the Figures.

9. We note you have included a table to which you do not refer in the text of your manuscript. Please ensure that you refer to Table 2 in your text; if accepted, production will need this reference to link the reader to the Tables.

Reviewers' comments:

Reviewer's Responses to Questions

**Comments to the Author**

1. Is the manuscript technically sound, and do the data support the conclusions?

Reviewer #1: Yes

Reviewer #2: Yes

2. Has the statistical analysis been performed appropriately and rigorously? 

Reviewer #1: Yes

Reviewer #2: Yes

3. Have the authors made all data underlying the findings in their manuscript fully available?

Reviewer #1: Yes

Reviewer #2: Yes

4. Is the manuscript presented in an intelligible fashion and written in standard English?

Reviewer #1: Yes

Reviewer #2: Yes

5. Review Comments to the Author

Reviewer #1: This manuscript investigates winter outdoor thermal comfort of older adults with different degrees of frailty in an urban park located in a hot-summer and cold-winter (HSCW) region of China. Overall, the topic is relevant, the dataset is relatively rich, and the methodology is generally appropriate.

1. In the Introduction, the authors need to more clearly emphasize how frailty-based classification differs from traditional age-based grouping and why it provides added scientific value for outdoor thermal comfort research.

2. In Section 2.3, please briefly explain how the FRAIL questionnaire was administered.

3. In Section 2.4, please specify the assumed values or ranges of the key personal parameters used in the RayMan software.

4. In the Results section, please specify the type of regression model used and report the relevant statistical indicators. A brief explanation of how the neutral temperature (NPET) was derived from the regression results would help improve clarity.

5. The finding in the Discussion that thermal sensitivity decreases with increasing frailty is interesting. However, The Discussion could benefit from a more explicit physiological or behavioral explanation. Please link the findings more clearly to existing indoor or medical studies on frailty and thermal perception.

Reviewer #2: The manuscript focuses on winter outdoor thermal comfort among older adults with different degrees of frailty in urban parks located in China's hot-summer and cold-winter (HSCW) region. The topic is timely and meaningful, given the rapid population aging and increasing attention to age-friendly urban design. The study presents a relatively complete research framework, clear methodological procedures, an adequate sample size, and appropriate statistical analyses. The results are presented in detail and discussed comprehensively. Overall, the manuscript falls within the scope of the journal and meets the basic academic standards.

However, several issues related to formatting consistency, literature updates, background justification, and graphical presentation should be carefully addressed to further improve the scientific rigor and publication quality of the manuscript. My specific comments are as follows:

1.Abbreviations and Formatting Issues

There are inconsistencies in the formatting of the Abbreviations section. Some full terms are capitalized while others are not; the formatting should be standardized throughout. In addition, widely recognized institutional abbreviations such as ASHRAE do not need to be listed in the abbreviations table.Abbreviations should be defined only at their first occurrence in the main text and subsequently used in their abbreviated form without repeated explanation (e.g., TSV and related indices).

Some equations and numbering formats appear inconsistent and should be revised according to the journal's formatting requirements.

2.Timeliness and Accuracy of References

Several references are relatively outdated (e.g., References 2, 3, and 6 are from around 2010). Considering the rapid development of outdoor thermal comfort research in recent years, the authors are encouraged to incorporate more recent studies (particularly from the last five years) to strengthen the manuscript's relevance and academic currency.

In addition, the journal name in Reference 2 is incorrect; it should be Ageing Research Reviews. The authors are advised to thoroughly check all references for accuracy, including journal titles, volume and issue numbers, page ranges, and formatting consistency.

3.Climate Zone Specification in the Title

The title would benefit from explicitly specifying the climate zone (i.e., hot-summer and cold-winter region, HSCW) to improve clarity and scientific precision.

Furthermore, the Introduction should provide a clearer justification for selecting Nanyang as the case study city. The authors should elaborate on its climatic representativeness, urban characteristics, or research gaps in existing studies to better justify the choice of study location.

4.Insufficient Climatic Context in the Introduction

The Introduction would be strengthened by including a more detailed description of Nanyang's climatic characteristics and comparing them with other cities in the HSCW region previously studied (e.g., Changsha, Wuhan, Chengdu). Highlighting similarities and differences would help clarify how this study complements or extends existing research and better demonstrate its contribution.

5.Figures and Graphical Clarity

In several figures, the axis labels and legends are too small, which affects readability. The font size should be enlarged to enhance clarity. The font style used in Figure 11 is inconsistent with the other figures and should be standardized.

The authors are encouraged to carefully review all figures to ensure consistent formatting, adequate resolution, uniform font style, appropriate line thickness, and clear color contrast. If necessary, the figures should be redrawn to improve overall visual quality and professionalism.

6. PLOS authors have the option to publish the peer review history of their article (what does this mean?). If published, this will include your full peer review and any attached files.

Reviewer #1: No

Reviewer #2: No

---

## [Author Response · Author response to Decision Letter 1]

11 Mar 2026

We appreciate the reviewers for the valuable comments. The revisions response can be found in the attached document "Response to Editor and Reviewers".

---

## [Decision Letter · Decision Letter 1]

31 Mar 2026

PONE-D-25-66927R1Winter outdoor thermal comfort of older people with different degrees of frailty: a study in Nanyang city, China's hot-summer and cold-winter regionPLOS One

Dear Dr. Biao Wang, Thank you for submitting your manuscript to PLOS ONE. After careful consideration, we feel that it has merit but does not fully meet PLOS ONE’s publication criteria as it currently stands. Therefore, we invite you to submit a revised version of the manuscript that addresses the points raised during the review process. Please submit your revised manuscript byMay 15 2026 11:59PM. If you will need more time than this to complete your revisions, please reply to this message or contact the journal office at plosone@plos.org. Please include the following items when submitting your revised manuscript:

We look forward to receiving your revised manuscript.

Kind regards,

Xuan Ma

Academic Editor

PLOS One

Journal Requirements:

Reviewers' comments:

Reviewer's Responses to Questions

**Comments to the Author**

1. If the authors have adequately addressed your comments raised in a previous round of review and you feel that this manuscript is now acceptable for publication, you may indicate that here to bypass the “Comments to the Author” section, enter your conflict of interest statement in the “Confidential to Editor” section, and submit your "Accept" recommendation.

Reviewer #1: All comments have been addressed

Reviewer #2: All comments have been addressed

2. Is the manuscript technically sound, and do the data support the conclusions?

Reviewer #1: Yes

Reviewer #2: Yes

3. Has the statistical analysis been performed appropriately and rigorously? 

Reviewer #1: Yes

Reviewer #2: Yes

4. Have the authors made all data underlying the findings in their manuscript fully available?

Reviewer #1: Yes

Reviewer #2: Yes

5. Is the manuscript presented in an intelligible fashion and written in standard English?

Reviewer #1: Yes

Reviewer #2: Yes

6. Review Comments to the Author

Reviewer #1: In Chapter 2.2, the measurement period appears limited (6 days), and representativeness is not discussed. Whether these days represent typical winter conditions and how measurement days were selected.

In Chapter 3.2, the preferred PET value for the frail group is unusually high for winter outdoor conditions, which raises concerns about interpretation. Please provide an explanation in the Discussion section

In Chapter 3.3, the manuscript switches from ordered regression to Spearman correlation without sufficient justification.

Reviewer #2: (No Response)

7. PLOS authors have the option to publish the peer review history of their article (what does this mean?). If published, this will include your full peer review and any attached files.

Reviewer #1: No

Reviewer #2: No

---

## [Author Response · Author response to Decision Letter 2]

6 Apr 2026

We appreciate the reviewers for the comments and suggestions. The revisions response can be found in the attached document "Response to Reviewers".

---

## [Decision Letter · Decision Letter 2]

5 May 2026

PONE-D-25-66927R2Winter outdoor thermal comfort of older people with different degrees of frailty: a study in Nanyang city, China's hot-summer and cold-winter regionPLOS One

Dear Dr. Wang,

Thank you for submitting your manuscript to PLOS ONE. After careful consideration, we feel that it has merit but does not fully meet PLOS ONE’s publication criteria as it currently stands. Therefore, we invite you to submit a revised version of the manuscript that addresses the points raised during the review process.

We look forward to receiving your revised manuscript.

Kind regards,

Xuan Ma

Academic Editor

PLOS One

**Journal Requirements:**

Reviewers' comments:

Reviewer's Responses to Questions

**Comments to the Author**

1. If the authors have adequately addressed your comments raised in a previous round of review and you feel that this manuscript is now acceptable for publication, you may indicate that here to bypass the “Comments to the Author” section, enter your conflict of interest statement in the “Confidential to Editor” section, and submit your "Accept" recommendation.

Reviewer #1: (No Response)

2. Is the manuscript technically sound, and do the data support the conclusions?

Reviewer #1: Partly

3. Has the statistical analysis been performed appropriately and rigorously? 

Reviewer #1: Yes

4. Have the authors made all data underlying the findings in their manuscript fully available?

Reviewer #1: Yes

5. Is the manuscript presented in an intelligible fashion and written in standard English?

Reviewer #1: Yes

6. Review Comments to the Author

**Reviewer #1:** The explanation attributes the higher preferred PET value (29.0 °C) to model extrapolation; however, the observed PET range falls significantly below this figure. Please clarify the validity and reliability of deriving a preferred value outside the range of the empirical data.

7. PLOS authors have the option to publish the peer review history of their article (what does this mean?). If published, this will include your full peer review and any attached files.

Reviewer #1: No

---

## [Author Response · Author response to Decision Letter 3]

7 May 2026

Please see the document named ''Response to Reviewers''.

---

## [Decision Letter · Decision Letter 3]

18 May 2026

Winter outdoor thermal comfort of older people with different degrees of frailty: a study in Nanyang city, China's hot-summer and cold-winter region

PONE-D-25-66927R3

Dear Dr. Biao Wang,

We’re pleased to inform you that your manuscript has been judged scientifically suitable for publication and will be formally accepted for publication once it meets all outstanding technical requirements.

Kind regards,

Xuan Ma

Academic Editor

PLOS One

Additional Editor Comments (optional):

Reviewers' comments:

Reviewer's Responses to Questions

**Comments to the Author**

1. If the authors have adequately addressed your comments raised in a previous round of review and you feel that this manuscript is now acceptable for publication, you may indicate that here to bypass the “Comments to the Author” section, enter your conflict of interest statement in the “Confidential to Editor” section, and submit your "Accept" recommendation.

Reviewer #1: All comments have been addressed

2. Is the manuscript technically sound, and do the data support the conclusions?

Reviewer #1: Yes

3. Has the statistical analysis been performed appropriately and rigorously? 

Reviewer #1: Yes

4. Have the authors made all data underlying the findings in their manuscript fully available?

Reviewer #1: Yes

5. Is the manuscript presented in an intelligible fashion and written in standard English?

Reviewer #1: Yes

6. Review Comments to the Author

Reviewer #1: (No Response)

7. PLOS authors have the option to publish the peer review history of their article (what does this mean?). If published, this will include your full peer review and any attached files.

Reviewer #1: No

---

## [Editor Report · Acceptance letter]

PONE-D-25-66927R3

PLOS One

Dear Dr. Wang,

I'm pleased to inform you that your manuscript has been deemed suitable for publication in PLOS One. Congratulations! Your manuscript is now being handed over to our production team.

Kind regards,

on behalf of

Professor Xuan Ma

Academic Editor

PLOS One